# Label Disentanglement in Partition-based Extreme Multilabel Classification

**Xuanqing Liu**[*]
Department of Computer Science
UCLA
xqliu@cs.ucla.edu

**Wei-Cheng Chang**
Amazon Inc.
weicheng.cmu@gmail.com

**Hsiang-Fu Yu**
Amazon Inc.
rofu.yu@gmail.com

**Cho-Jui Hsieh**
Department of Computer Science
UCLA
chohsieh@cs.ucla.edu

**Inderjit Dhillon**
Amazon Inc. & UT Austin
inderjit@cs.utexas.edu

## Abstract

Partition-based methods are increasingly-used in extreme multi-label classification (XMC) problems due to their scalability to large output spaces (e.g., millions or more). However, existing methods partition the large label space into mutually exclusive clusters, which is sub-optimal when labels have multi-modality and rich semantics. For instance, the label "Apple" can be the fruit or the brand name, which leads to the following research question: can we disentangle these multi-modal labels with non-exclusive clustering tailored for downstream XMC tasks? In this paper, we show that the label assignment problem in partition-based XMC can be formulated as an optimization problem, with the objective of maximizing precision rates. This leads to an efficient algorithm to form flexible and overlapped label clusters, and a method that can alternatively optimizes the cluster assignments and the model parameters for partition-based XMC. Experimental results on synthetic and real datasets show that our method can successfully disentangle multi-modal labels, leading to state-of-the-art (SOTA) results on four XMC benchmarks.

## 1 Introduction

The eXtreme Multi-label Classification (XMC) task is to find relevant labels from an enormous output space of candidate labels, where the size is in the order of millions or more ([1, 2, 3, 4, 5]; etc.) This problem is of interest in both academia and industry: for instance, tagging a web page given its contents from tens of thousands of categories [6]; finding a few products that a customer will purchase from among an enormous catalog on online retail stores [7]; or recommending profitable keywords given an item/product from millions of advertisement keywords [8].

The XMC problem is challenging not only because of the data scalability (e.g., the number of instances, features, and labels are of the scale of millions or more), but also due to the label sparsity issue where there is little training signal for long-tailed labels. To tackle these issues, most prevailing XMC algorithms use a *partition-based* approach. Instead of ranking the entire set of millions of labels, they partition the label space into smaller mutually-exclusive clusters. Each instance is only mapped to one or a few label clusters based on a matching model, and then ranking is only conducted within the smaller subset of labels. Some exemplars are Parabel [8], eXtremeText [9], AttentionXML [10], XR-Linear [11] and X-Transformer [12].

---

[*]Work done while Xuanqing was interning at Amazon Inc. Code will be released at `https://github.com/amzn/pecos`.

35th Conference on Neural Information Processing Systems (NeurIPS 2021).

Partitioning labels into mutually-exclusive clusters may not be ideal. When labels are semantically complex and multi-modal, it is more natural to assign a label to multiple semantic clusters. In product categorization, for instance, the tag "belt" can be related to a vehicle belt (under "vehicle accessories" category), or a man's belt (under "clothing" category). Assigning "belt" to just one of the clusters but not the other is likely to cause a mismatch for certain queries. To solve this problem, we reformulate the label clustering step as an assignment problem, where each label can be assigned to multiple clusters to allow disentanglement of mixed semantics. Further, we formulate learning optimal assignments by maximizing the precision as an optimization problem, and propose efficient solvers that automatically learn a good label assignment based on the current matching models. With this formulation, we apply our algorithm to alternatively refine label assignments and matching model in existing partition-based XMC methods to boost their performance. Similar idea can be found in [13, 14], however, our tree structures are constructed in one pass with hierarchical $k$-means. Our contributions can be summarized below:

- We propose a novel way to obtain label assignments that disentangle multi-modal labels to multiple clusters.

- Unlike previous methods that partition the label set before training, we propose an optimization-based framework that allows optimizing label assignment with the matching and ranking modules inside a partition-based XMC solver. Our method is plug-and-play; it is orthogonal to the models so most of the partition-based methods can benefit from our method.

- Our proposed solution yields consistent improvements over two leading partition-based methods, XR-Linear [11] and X-Transformer [12]. Notably, with the concatenation of tfidf features and X-Transformer embeddings, we achieve new SOTA results on four XMC benchmark datasets.

## 2 Related Work

### 2.1 XMC literature

**Sparse Linear Models** Sparse linear one-versus-rest (OVR) methods such as DiSMEC [4], ProXML [15], PDSparse [16], PPDSparse [17] explore parallelism to speed up the algorithm and reduce the model size by truncating model weights to encourage sparsity. OVR approaches are also building blocks for many other XMC approaches. For example, in Parabel [8], SLICE [18], X-Transformer [12], linear OVR classifiers with negative sampling are used.

**Partition-based Methods** The efficiency and scalability of sparse linear models can be further improved by incorporating different partitioning techniques on the label spaces. For instance, Parabel [8] partitions the labels through a balanced 2-means label tree using label features constructed from the instances. Other approaches attempt to improve on Parabel, for instance, eXtremeText [9], Bonsai [19], NAPKINXC [20], and XR-Linear [11] relax two main constraints in Parabel by: 1) allowing multi-way instead of binary partitions of the label set at each intermediate node, and 2) removing strict balancing constraints on the partitions. More recently, AttentionXML [10] uses BiLSTMs and label-aware attention to replace the linear functions in Parabel, and warm-up training the models with hierarchical label trees. In addition, AttentionXML considers various negative sampling strategies on the label space to avoid back-propagating the entire bottleneck classifier layer.

**Graph-based Methods** SLICE [18] uses an approximate nearest neighbor (ANN) graph as an indexing structure over the labels. For a given instance, the relevant labels can be found quickly via ANN search. SLICE then trains linear OVR classifiers with negative samples induced from ANN. Graph-based partitions can be viewed as an extension of tree-based partitions, where at each layer of the tree, random edges are allowed to connect two leaf nodes to further improve connectivity. Nevertheless, such construction of overlapping tree structures is fully unsupervised, and is agnostic to the data distribution or training signals of the downstream XMC problem.

### 2.2 Overlapped Clustering

Finding overlapped clusters has been studied in the unsupervised learning literature [21, 22, 23, 24]. For example, Cleuziou *et al.* [22] extend $K$-means with multi-assignments based on coverage, but

tends to yield imbalanced clusters. This issue was later improved by Lu *et al.* [23] with sparsity constraints. Recently, Whang *et al.* [24] propose variants of $k$-mean objectives with flexible clustering assignment constraints to trade-off the degree of overlapping. However, all these unsupervised approach cannot be optimized with existing partition-based XMC methods.

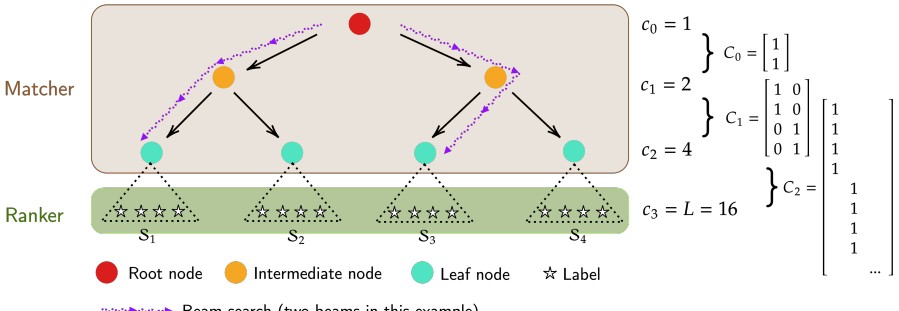

Figure 1: An illustration of partition-based XMC models. The matcher is a tree structured model similar to Parabel or XR-Linear. In this diagram, the hierarchical label tree has a depth of 3, branching factor of 2, number of clusters $K = 4$, and number of labels $L = 16$.

## 3 Background of partition-based XMC

The XMC problem can be characterized as follows: given an input instance $\boldsymbol{x} \in \mathbb{R}^d$ and a set of labels $\mathbb{L} = \{1, 2, \ldots, L\}$, find a model that retrieves top relevant labels in $\mathbb{L}$ efficiently. The model parameters are estimated from the training dataset $\{(\boldsymbol{x}_i, \boldsymbol{y}_i) : i = 1, \ldots, n\}$ where $\boldsymbol{y}_i \in \{0, 1\}^L$ denotes the relevant labels for $\boldsymbol{x}_i$ from the label space $\mathbb{L}$. We further denote $\boldsymbol{X} \in \mathbb{R}^{n \times d}$ as the feature matrix and $\boldsymbol{Y} \in \mathbb{R}^{n \times L}$ as the label matrix.

Partition-based XMC methods rely on label space partitioning to screen out irrelevant labels before running the ranking algorithm. They often have the following three components (depicted in Figure 1):

- The **cluster assignments**, where a set of labels are assigned to each cluster. Assuming there are $K$ clusters, we use $\mathcal{S}_i = \{\ell_1^i, \ell_2^i, \ldots\}$ to denote the labels assigned to cluster $i$, and $\bigcup_{i=1}^{K} \mathcal{S}_i = \mathbb{L}$. The clusters are constructed by $K$-means on label features, such that the clusters $\{\mathcal{S}_i\}_{i=1}^{K}$ are mutually exclusive and unaware of the matcher $\mathcal{M}$ and ranker $\mathcal{R}$, since clustering is typically performed only once *before* the matcher and ranker are trained. On the contrary, as we will see shortly, our new method provides a way to refine cluster assignments based on the matcher and is able to effectively unveil multimodal labels.

- The **matcher**, which matches the input data $\boldsymbol{x}_i \in \mathbb{R}^d$ to a small set of candidate label clusters

$$\mathcal{M} : \boldsymbol{x}_i \mapsto \{\mathcal{S}_1, \mathcal{S}_2, \ldots, \mathcal{S}_b\}, \quad b \leq K. \tag{1}$$

Each label set $\mathcal{S}_k$ contains a small fraction of labels (a few hundreds), and $b$ is called the beam size. One recursive implementation of matcher is seen in XR-Linear [11], where inside the matcher, there is a tree constructed by recursive $K$-means clustering. On each level of the tree, a maximum of $b$ nodes is selected according to the scores obtained from linear models. After that, all siblings of $b$ nodes will be expanded at the next level to repeat this process recursively until $b$ leaves are obtained, which results in the match set $\{\mathcal{S}_1, \mathcal{S}_2, \ldots, \mathcal{S}_b\}$. See the "Matcher" block in Figure 1.

- The **ranker**, which ranks the candidate labels collected from the matcher, with $\succ$ denoting the ranking order:

$$\mathcal{R} : \mathcal{M}(\boldsymbol{x}_i) \mapsto \ell_{(1)} \succ \ell_{(2)} \succ \cdots \succ \ell_{(w)},$$
$$\text{where} \quad \{\ell_{(1)}, \ell_{(2)}, \ldots, \ell_{(w)}\} = \bigcup_{i=1}^{b} \mathcal{S}_i. \tag{2}$$

Lastly, top-$k$ labels are returned as the final prediction. See the "Ranker" block of Figure 1.

Partition-based XMC typically assumes labels in the same cluster are similar, and thus when training the ranker, they focus on distinguishing the labels (and the corresponding samples) within each cluster.

Taking the widely used linear ranker as an example, they often assign a weight vector for each label. The weight vector $\boldsymbol{w}_\ell$ for label $\ell$ can be obtained by the following training objective:

$$\min_{\boldsymbol{w}_\ell} \sum_{i \in \mathcal{D}_{\text{positive}}} \mathcal{L}(\boldsymbol{x}_i^\top \boldsymbol{w}_\ell; +1) + \sum_{i \in \mathcal{D}_{\text{negative}}} \mathcal{L}(\boldsymbol{x}_i^\top \boldsymbol{w}_\ell; -1), \tag{3}$$

where $\mathcal{L}(\cdot, \cdot)$ is the loss function (e.g., squared-hinge loss); $\mathcal{D}_{\text{positive}}$ and $\mathcal{D}_{\text{negative}}$ are the positive and negative samples for label $\ell$. In XMC, it is not efficient to collect all non-positive data $\mathcal{D} \setminus \mathcal{D}_{\text{positive}}$ as the negative part to train $\boldsymbol{w}_\ell$; instead, they often sample the "hard negatives" which is a tiny subset $\mathcal{D}_{\text{negative}} \subset \mathcal{D} \setminus \mathcal{D}_{\text{positive}}$. Different negative sampling changes the loss function and the final results. For instance, in Parabel [8] and XR-Linear [11], $\mathcal{D}_{\text{negative}}$ are the examples having similar labels but not $\ell$ itself, based on the intuition that the learning can be more efficient by separating similar but different labels. In other words,

$$\mathcal{D}_{\text{positive}} = \{\boldsymbol{x}_i \mid \ell \in \boldsymbol{y}_i\}, \quad \mathcal{D}_{\text{negative}} = \{\boldsymbol{x}_i \mid \ell \notin \boldsymbol{y}_i \text{ but "similar"}\}. \tag{4}$$

The label similarity information is hidden in the cluster assignments $\{\mathcal{S}_1, \mathcal{S}_1, \ldots, \mathcal{S}_K\}$, with the assumption that similar labels are clustered.

## 4 Proposed Method

### 4.1 Motivation

The central idea of the partition-based XMC methods is to partition labels into disjoint clusters, such that labels in each cluster share similar semantics. This relies on the *unimodal assumption* that each label only represents a pure, uniform semantic across all positive samples. However, we observe in practice that this assumption may not hold in general. Given labels that have multi-modal semantics, it is natural to seek a method to *disentangle* their semantics from each other and treat each semantic as a separate label, and further assign them to different clusters. To achieve this, we weaken the requirement that label clusters under leaf nodes are mutually exclusive $\mathcal{S}_i \cap \mathcal{S}_j = \varnothing$, to a more suitable, limited overlapping: for any label $\ell \in \{1, 2, \ldots, L\}$, it can appear *at most* $\lambda$-times among $\{\mathcal{S}_1, \mathcal{S}_2, \ldots, \mathcal{S}_K\}$.

Although allowing label clusters to overlap with each other does not explicitly disentangle the semantics of a label, it paves the way for learning multiple versions of weights for the same label. Take the linear model in Eq. (3) for simplicity, label $\ell$ is seen in both cluster $\mathcal{S}_A$ and $\mathcal{S}_B$, so we will have two weight vectors $\boldsymbol{w}_A$ and $\boldsymbol{w}_B$ for the same label. $\boldsymbol{w}_A$ and $\boldsymbol{w}_B$ are trained separately in Eq. (3), where they have different negative examples $\mathcal{D}_{\text{negative}}^A$ and $\mathcal{D}_{\text{negative}}^B$. Recall in Eq. (4) the negative part of the data relies on the label similarity induced from clusters $\mathcal{S}_A$ and $\mathcal{S}_B$. Therefore, $\boldsymbol{w}_A$ and $\boldsymbol{w}_B$ will eventually converge to two different solutions – this is how the semantics are disentangled when we assign a label to multiple clusters.

### 4.2 Optimization-based Label Assignment Approaches

How to assign a label to multiple clusters? Previous methods consider clustering as a preprocessing step and apply unsupervised methods, such as $K$-means, to partition the label space. In contrast, we formulate label assignments as an optimization problem to maximize the precision rate for XMC, which allows learning a matcher-aware clustering assignment as well as alternative updates between cluster assignments and the matcher.

As mentioned before, our goal is to learn $\mathcal{S}_1, \ldots, \mathcal{S}_K$ to cover all the labels, and the sets can be overlapping. To find the best label assignments, it is natural to maximize the precision rate. Many previous methods in information retrieval obtain ranking functions by optimizing (surrogates) Precision/Recall [25, 26]. Here our goal is to find a good combination of **matcher** and **cluster assignments** in the context of partition-based XMC, so the objective is totally different.

To formally define precision, we first define the output of matcher represented by matrix $\boldsymbol{M} \in \mathbb{R}^{n \times K}$:

$$\boldsymbol{M}_{ij} = \begin{cases} 1, & \text{if } \boldsymbol{x}_i \text{ is matched to leaf cluster } S_j, \\ 0, & \text{otherwise.} \end{cases} \tag{5}$$

For beam size $b$, every row of $\boldsymbol{M}$ has exactly $b$ nonzero entries. At the same time, the cluster assignments $\mathcal{S}_1, \ldots, \mathcal{S}_K$ can be parameterized by the clustering matrix $\boldsymbol{C} \in \{0, 1\}^{L \times K}$ so that

$C_{ij} = 1$ if and only if label-$i$ is seen in cluster $\mathcal{S}_j$, otherwise $C_{ij} = 0$. With this notation, the candidate labels generated by matcher is given by $\hat{Y} := \text{Binary}(MC^\top), \hat{Y} \in \mathbb{R}^{n \times L}$, where $\text{Binary}(A) = I_A$ is the element-wise indicator function.

Now consider the number of true positives in top-$k$ predictions (TP@$k$), it is upper bounded by the intersection between matcher predictions $\hat{Y}$ and the ground truth $Y$, i.e.

$$\text{TP@}k = |\text{Top-}k(\hat{Y}) \odot Y| \overset{(i)}{\leq} |\hat{Y} \odot Y| \overset{(ii)}{=} \text{Tr}(Y^\top \hat{Y}), \tag{6}$$

where $|\cdot|$ denotes the #nnz elements in a sparse matrix; the inequality $\overset{(i)}{\leq}$ is due to the error of ranker hidden in Top-$k(\cdot)$; and $\overset{(ii)}{=}$ is from the fact that both $\hat{Y}$ and $Y$ are binary.

In this paper, we consider following scenarios that simplifies our analysis

- Perfect ranker: the ranker does not make any mistake, so the gap in Eq. (6) vanishes.

- Probably correct ranker: for any true positive label $\ell^+ \in \hat{Y}$, it is ranked to top-$k$ with a *constant* probability $p$, i.e. $\mathbb{P}(\ell^+ \in \text{Top-}k(\hat{Y})|\ell^+ \in \hat{Y}) = p$. Then we have $\text{TP@}k = p \cdot \text{Tr}(Y^\top \hat{Y})$.

In both cases, the precision is proportional to $\text{Tr}(Y^\top \hat{Y})$. We can then formulate the problem of learning the best cluster assignment as

$$\underset{\substack{\mathcal{S}_1, \mathcal{S}_2, \ldots, \mathcal{S}_K \\ \cup_{i=1}^{K} \mathcal{S}_i = \{1,2,\ldots,L\}}}{\text{maximize}} \text{Tr}(Y^\top \text{Binary}(MC^\top)), \quad \text{s.t.} \sum_{i=1}^{K} I_{\ell \in \mathcal{S}_i} \leq \lambda, \forall \ell \in \{1,2,\ldots,L\}. \tag{7}$$

Note that $\{\mathcal{S}_i, \ldots, \mathcal{S}_K\}$ are hidden in $C$. The first constraint ensures we cover the whole label set and the second constraint mitigates the case of degenerated clusters where some $\mathcal{S}_i$'s have too many labels, resulting in significantly increased time complexity. The parameter $\lambda$ is the only hyperparameter to be tuned. In practice, we find $\lambda$ is stable across datasets and for simplicity we select $\lambda = 2$ for all the experiments. We also show the sensitivity of our algorithm with respect to $\lambda$ in Section 5.2.

The optimization problem (7) is combinatorial and hard to solve. In fact, we show this is NP-complete:

**Theorem 1.** *Problem* (7) *is NP-complete.*

This can be proved by a polynomial time reduction from the set cover problem to Problem (7), and the proof is deferred to the Appendix A.1. To develop an efficient algorithm, we approximate the objective function of (7) with a continuous, ReLU-like function

$$\text{Binary}(MC^\top) \approx \max(MC^\top, 0) = MC^\top, \tag{8}$$

where the first approximation comes from replacing binary function to ReLU; the second equality is because both $M$ and $C$ are positive in all entries. A special case is when beam size $b = 1$. In that case $MC^\top$ is a binary matrix, so $\text{Binary}(MC^\top) = MC^\top$. We then consider the following simplified problem:

$$\underset{\substack{\mathcal{S}_1, \mathcal{S}_2, \ldots, \mathcal{S}_K \\ \cup_{i=1}^{K} \mathcal{S}_i = \{1,2,\ldots,L\}}}{\text{maximize}} \text{Tr}(Y^\top MC^\top), \quad \text{s.t.} \sum_{i=1}^{K} I_{\ell \in \mathcal{S}_i} \leq \lambda, \forall \ell \in \{1,2,\ldots,L\}. \tag{9}$$

This problem can be solved efficiently with a closed form solution:

**Theorem 2.** *Problem* (9) *has a closed form solution*

$$C^* = Proj(Y^\top M), \tag{10}$$

*where the $Proj(\cdot)$ operator selects the top-$\lambda$ elements for each row of the matrix.*

The above result can be easily derived since the objective function is linear and the constraint is a row-wise $\ell_0$ norm constraint. The detailed proof can be found in the Appendix A.2.

But is (10) also a good solution to the original problem (7)? In fact, we show that this solution, despite not being optimal for (7), is provably better than any non-overlapping cluster partitioning, which is used in almost all the existing partition-based XMC methods.

**Theorem 3.** *For any clustering matrix $C$ corresponding to a non-overlapping partition of the label set, we have* $\text{Tr}(Y^\top Binary(M(C^*)^\top)) > \text{Tr}(Y^\top Binary(MC^\top))$.

The proof can be found in the Appendix A.3. This theorem implies our partitioning can achieve higher precision over any existing non-overlapping clustering for any ranker.

**Practical Implementation**    After $C$ (or equivalently $\{\mathcal{S}_i\}_{i=1}^{K}$) is determined, we finetune the matcher $\mathcal{M}$ to accommodate the new deployment of label clusters. Note that most of the existing partition-based XMC starts from unsupervised label clustering and then alternate between getting a new (overlapping) clustering and finetune the matcher. Our algorithm can plug-in into most of these methods by adding one or more loops of alternative cluster assignments and matcher updates. Given the current matcher, the cluster assignments are updated based on the proposed formulation (10), and then the matcher will be retrained following the same procedure used in the original partition-based XMC solver. The whole routine is exhibited in Algorithm 1. Since we use a balanced $K$-means label clustering as initialization, we found that after one step update in Line 6 of Algorithm 1, $\{\mathcal{S}_i\}_{i=1}^{K}$ is still not too imbalanced in our experiments. It's possible to add cluster size constraints in Problem (9) to enforce the label partition to be balanced, as discussed in Appendix A.4, but we do not find a practical need for such constraints.

---

**Algorithm 1** Our proposed framework.

---

1: **Input:** training data $\langle \boldsymbol{X}, \boldsymbol{Y} \rangle$, any partition-based XMC algorithm `XMC-part`.
2: **Output:** the trained model (i.e., matcher $\mathcal{M}$, ranker $\mathcal{R}$, label clusters $\{\mathcal{S}_i\}_{i=1}^{K}$).
3: **Initialize** $\{\mathcal{S}_i\}_{i=1}^{K}$ with balanced $K$-means using label features.
4: **Initialize** $\mathcal{M} \leftarrow \texttt{XMC-part}(\{\mathcal{S}_i\}_{i=1}^{K})$
5: Compute matcher prediction matrix $\boldsymbol{M} = \mathcal{M}(\boldsymbol{X})$ by Eq. (5).
6: Update label clustering: $\{\mathcal{S}_i\}_{i=1}^{K}$ by Eq. (10).
7: *%% Following two lines are called "alternative update" hereafter.*
8: Finetune the matcher given new clusters: $\mathcal{M} \leftarrow \texttt{XMC-part}(\{\mathcal{S}_i\}_{i=1}^{K})$.
9: Train the ranker with updated clusters and matcher: $\mathcal{R} \leftarrow \texttt{XMC-part}(\{\mathcal{S}_i\}_{i=1}^{K}, \mathcal{M})$.

---

**Deduplication at inference time.** To perform inference with the new model, we need to deduplicate the scores from the same labels but in different clusters. This happens because we use beam search to efficiently search $b$ paths from root node to leaves. Refer to Figure 1 where $b = 2$ is shown ($\mathcal{S}_1$ and $\mathcal{S}_3$ selected), if both $\mathcal{S}_1$ and $\mathcal{S}_3$ contain a same label $\ell_{\text{duplicate}}$ but different scores $s_1$ and $s_3$, then we average them together with the final score $s(\ell_{\text{duplicate}}) = \frac{1}{2}(s_1 + s_3)$. In this sense, our algorithm can be interpreted as a finer-grained ensemble which ensembles the scores inside a tree, rather than ensembling multiple, independently trained trees.

## 5    Experimental Results

Our proposed framework serves as a *generic plugin* for any partition-based XMC methods and we show its efficacy on multiple experiment setups. First, we verify that the proposed method improves over the baselines on synthetic datasets where we simulate labels with mixed semantics. Next, we test the sensitivity of hyperparameter $\lambda$, followed by experiments on real-world XMC benchmark datasets. We end this section with an ablation study.

**Datasets.** We consider four publicly available XMC benchmark datasets [2, 10] for our experiments. See Table 1 for data statistics. To obtain state-of-the-art (SOTA) results in Table 3, we concatenate the dense neural embeddings (from fine-tuned X-Transformer [12, 11]) and sparse TF-IDF features (from AttentionXML [10]) as the input features to train the models.

**Evaluation Metric.** We measure the performance with precision metrics (P@k) as well as Propensity-based scores (PSP@k) [5], which are widely-used in the XMC literature [27, 8, 28, 18, 12, 11]. Specifically, for a predicted score vector $\hat{\mathbf{y}} \in \mathbb{R}^L$ and a ground truth label vector $\mathbf{y} \in \{0, 1\}^L$, $\text{P@}k = \frac{1}{k} \sum_{l \in \text{rank}_k(\hat{\mathbf{y}})} \mathbf{y}_l$; $\text{PSP@}k = \frac{1}{k} \sum_{l=1}^{k} \frac{\mathbf{y}_{\text{rank}(l)}}{\mathbf{p}_{\text{rank}(l)}}$, the latter focuses more on the *tail labels*.

**Models.** As we introduce a new technique that is generally applicable, our method must be combined with existing partition-based XMC algorithms. We take XR-Linear [11] as the backbone for all experiments except Section 5.3. To get SOTA results in large-scale real datasets, we change the backbone to X-Transformer [12] in Section 5.3. The implementation details for combining our method with XR-Linear and X-Transformer are discussed in the Appendix A.5.

**Hyper-parameters.** Our technique depends on hyperparameter $\lambda$, which is tested in a standalone section. For hyperparameters in the XMC model, we mostly follow the default settings in the corresponding software. The details about hyperparameters are listed in the Appendix A.6.

| Dataset | $n_{\text{trn}}$ | $n_{\text{tst}}$ | $L$ | $\bar{L}$ | $\bar{n}$ | $d_{\text{tfidf}}$ |
|---|---|---|---|---|---|---|
| Wiki10-31K | 14,146 | 6,616 | 30,938 | 18.64 | 8.52 | 101,938 |
| AmazonCat-13K | 1,186,239 | 306,782 | 13,330 | 5.04 | 448.57 | 203,882 |
| Amazon-670K | 490,449 | 153,025 | 670,091 | 5.45 | 3.99 | 135,909 |
| Amazon-3M | 1,717,899 | 742,507 | 2,812,281 | 36.04 | 22.02 | 337,067 |

Table 1: XMC data statistics. $n_{\text{trn}}$ and $n_{\text{tst}}$ are the number of instances in training and testing splits. $L$ is the number of labels and $\bar{L}$ is the average number of labels in each instance. $\bar{n}$ is the average number of positive data of each label. $d_{\text{tfidf}}$ is the sparse TFIDF feature dimension. These four datasets and the sparse TF-IDF features are downloaded from `https://github.com/yourh/AttentionXML` which are the same as used in AttentionXML [10] and X-Transformer [12].

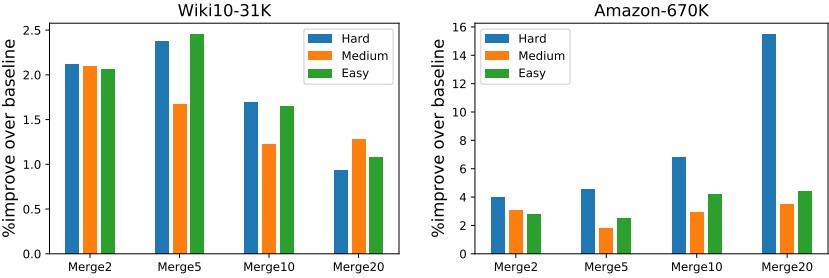

Figure 2: The relative precision@5 gain over baseline XR-Linear model when adding our proposed overlapping clusters. The precision values are shown in the Appendix A.7

## 5.1 Synthetic datasets by grouping true labels

In this experiment, we create a synthetic data derived from public datasets by deliberately entangling some labels. The motivation is to see the capability of disentangling different semantics from fused labels. Specifically, we designed several methods to mix up labels in the order of easy, medium and hard. The grouping is done without replacement: suppose a dataset has $L$ labels, and we want to entangle every $k$ labels into a fake one, then after this process we will get an artificial dataset with $\lceil \frac{L}{k} \rceil$ labels. Below are three implementations we are going to evaluate:

- Easy mode: we run a balanced $\lceil \frac{L}{k} \rceil$-means clustering on the original label space of size $L$, so that each cluster contains about $k$ labels. After that, we represent each cluster with a separate label, and the new label embedding is the cluster centroid.

- Medium mode: we run a balanced $\lceil \frac{L}{32k} \rceil$-means clustering on the label space. Different from the setting above, now each cluster contains $32k$ labels. Next, we *randomly* group the $32k$ labels in to 32 subsets, with $k$ labels each.

- Hard mode: the most difficult mode is randomly shuffle labels into $\lceil \frac{L}{k} \rceil$ subsets, with $k$ labels in each subset.

We label the difficulty of three modes based on the idea that when similar labels are entangled together, it is less a issue for the baseline model, so the accuracy drop is negligible; Whereas if unrelated labels are bundled together, our method starts to shine. See Figure 2 for the relative performance improvement over XR-Linear. Several conclusions can be made from this figure: first, the improvements are all positive, meaning our method consistently beats the baseline XR-Linear model in all scenarios; secondly, by comparing Wiki10-31K with Amazon-670K, we see Amazon-670K shows an upwards in performance gain. This is mainly because the larger label space of Amazon-670K renders this problem even harder after combining labels, so our method exhibits a big edge over baselines. How are the labels disentangled to different clusters? We attach a random example from our dataset in Table 2. This example shows that our method can indeed separate labels with fused semantics, improving the quality of both matcher and ranker.

| Methods | P@1 | P@3 | P@5 | PSP@1 | PSP@3 | PSP@5 | P@1 | P@3 | P@5 | PSP@1 | PSP@3 | PSP@5 |
|---|---|---|---|---|---|---|---|---|---|---|---|---|
| | Wiki10-31K | | | | | | AmazonCat-13K | | | | | |
| AnnexML [3] | 86.46 | 74.28 | 64.20 | 11.86 | 12.75 | 13.57 | 93.54 | 78.36 | 63.30 | 51.02 | 65.57 | 70.13 |
| DiSMEC [4] | 84.13 | 74.72 | 65.94 | 10.60 | 12.37 | 13.61 | 93.81 | 79.08 | 64.06 | 51.41 | 61.02 | 65.86 |
| Parabel [8] | 84.19 | 72.46 | 63.37 | 11.69 | 12.47 | 13.14 | 93.02 | 79.14 | 64.51 | 50.92 | 64.00 | 72.10 |
| eXtremeText [9] | 83.66 | 73.28 | 64.51 | - | - | - | 92.50 | 78.12 | 63.51 | - | - | - |
| Bonsai [19] | 84.52 | 73.76 | 64.69 | 11.85 | 13.44 | 14.75 | 92.98 | 79.13 | 64.46 | 51.30 | 64.60 | 72.48 |
| XML-CNN [27] | 81.41 | 66.23 | 56.11 | 9.39 | 10.00 | 10.20 | 93.26 | 77.06 | 61.40 | 52.42 | 62.83 | 67.10 |
| AttentionXML [10] | 87.47 | 78.48 | 69.37 | 15.57 | 16.80 | 17.82 | 95.92 | 82.41 | 67.31 | 53.76 | 68.72 | 76.38 |
| XR-LINEAR [11] | 85.13 | 74.96 | 66.05 | - | - | - | 94.54 | 79.87 | 64.67 | - | - | - |
| X-Transformer [11] | 88.26 | 78.51 | 69.68 | 15.12 | 16.52 | 17.99 | 96.48 | 83.41 | 68.19 | 50.36 | 66.32 | 76.45 |
| Ours | 88.85 | 79.52 | 70.67 | 15.23 | 16.81 | 18.42 | 96.48 | 83.55 | 68.36 | 50.46 | 66.76 | 77.01 |
| | Amazon-670K | | | | | | Amazon-3M | | | | | |
| AnnexML [3] | 42.09 | 36.61 | 32.75 | 21.46 | 24.67 | 27.53 | 49.30 | 45.55 | 43.11 | 11.69 | 14.07 | 15.98 |
| DiSMEC [4] | 44.78 | 39.72 | 36.17 | 26.26 | 30.14 | 33.89 | 47.34 | 44.96 | 42.80 | - | - | - |
| Parabel [8] | 44.91 | 39.77 | 35.98 | 26.36 | 29.95 | 33.17 | 47.42 | 44.66 | 42.55 | 12.80 | 15.50 | 17.55 |
| eXtremeText [9] | 42.54 | 37.93 | 34.63 | - | - | - | 42.20 | 39.28 | 37.24 | - | - | - |
| Bonsai [19] | 45.58 | 40.39 | 36.60 | 27.08 | 30.79 | 34.11 | 48.45 | 45.65 | 43.49 | 13.79 | 16.71 | 18.87 |
| XML-CNN [27] | 33.41 | 30.00 | 27.42 | 17.43 | 21.66 | 24.42 | - | - | - | - | - | - |
| AttentionXML [10] | 47.58 | 42.61 | 38.92 | 30.29 | 33.85 | 37.13 | 50.86 | 48.04 | 45.83 | 15.52 | 18.45 | 20.60 |
| XR-LINEAR [11] | 42.51 | 37.32 | 33.60 | - | - | - | 46.65 | 43.38 | 41.05 | - | - | - |
| X-Transformer [12, 11] | 48.07 | 42.96 | 39.12 | 36.06 | 38.38 | 41.04 | 51.20 | 47.81 | 45.07 | 18.64 | 21.56 | 23.65 |
| Ours | 50.70 | 45.40 | 41.55 | 36.39 | 39.15 | 41.96 | 52.70 | 49.92 | 47.71 | 18.79 | 21.90 | 24.10 |

Table 3: Comparing the Precision@k (P@k) and Propensity-based scores (PSP@k) for $k = 1, 3, 5$ on four datasets. First place is marked red; second place is marked blue. Our method is trained with same concatenation of dense neural and sparse TF-IDF features, where the former is from fine-tuned X-Transformer [12, 11]. Used as a plugin upon existing X-Transformer models, our proposed framework achieves new SOTA results on three out of four datasets.

## 5.2 Hyperparameter sensitivity ($\lambda$)

In real applications, we never know the number of semantics a label has - either because the label space is huge, or it is uneconomical. In this section, we check the sensitivity of model performance on hyperparameter $\lambda$. To this end, we consider four different datasets and choose $\lambda = \{1, 2, \ldots, 6\}$. The results are exhibited in Figure 3. From the figure above, we observe that the performance generally improves as we increase $\lambda$ – our prior on the number of different meanings of each labels. However, since the benefits diminishes quickly, we choose $\lambda = 2$ for all the following experiments in this paper.

| Fused label: "tanks" and "fashion scarves" | |
|---|---|
| Cluster ID | Input text |
| 96 | Chiffon Butterfly Print on Black - Silk Long Scarf 21x; (Clearance): ... offers you one of delightful styles as you desire. |
| 103 | This Scuba Yoke Fill Station device allows a person to fill high pressure nitrogen/compressed air tanks from a scuba tank... |

Table 2: An example of how a fused label is disentangled into two labels, both then falling into different clusters (cluster ids are 96 and 103), finally labeled correctly despite the fusion of labels.

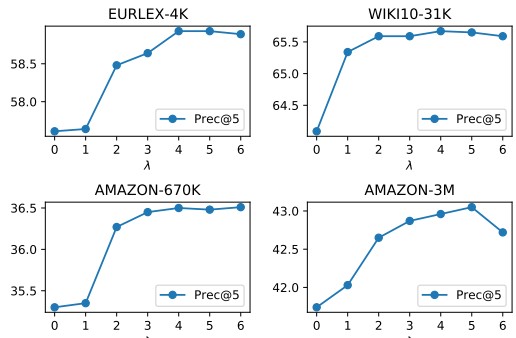

Figure 3: Precision@5 (y-axis) versus different $\lambda$ ranging from 0 to 6. We generally see that higher $\lambda$ converts to better performance, but it plateaus at $\lambda = 4$.

## 5.3 Real datasets

The above experiments showed that our method is indeed better at disentangling complex meanings from labels. In real scenarios, it is certainly not the case that every label will have tens of underlying semantics. On the contrary, they behave more like a mix of both unimodal and multimodal labels. However, in this experiment, we will show that our method is still favorable. We conduct experiments to show that our method can successfully boost the performance of existing partition-based XMC models, including XR-Linear [11] and X-Transformer [11].

In Table 4, we combine the proposed method with competitive linear models XR-Linear on four XMC benchmark datasets. We assume only the TF-IDF features are given and assume the single model setting without any ensemble. Our method consistently improves the performance over the baseline XR-Linear. Moreover, we tested how finetuning increases the accuracies (by comparing No/Yes in "alternative update?" column). The results prove that it is indeed necessary to train both the model parameters and cluster map in an alternative fashion (i.e., see Algorithm 1).

Next, we repeat the same process but based on the current SOTA X-Transformer model. The results are shown in Table 3. Notice that our method reuses the same dense+sparse features as in the X-Transformer model, and the results are the ensemble of 9 models [12]. From Table 3 we can observe that our method achieves new SOTA precision results with a significant margin while still maintaining good PSP scores, comparing with other strong algorithms such as Parabel, AttentionXML, XR-Linear and X-Transformer. Notably, the numbers show that our method is more suitable for large label space (such as Amazon-3M) compared with smaller label space (e.g., AmazonCat-13K).

To benchmark the training and inference overhead due to repeated clustering and training in our method, we report the relative training and inference time under different $\lambda$'s shown in Table 5. We remark that our method is more economical when the underlying model is inherently slow to train. On Amazon-3M for example, our framework only introduces 1.1% training overhead and 21.1% inference overhead over the X-Transformer models.

In Appendix A.8, we provided one more experiment on the real-world datasets that compares our method with random baseline. The random baseline is constructed by duplicating the same amount of labels to random clusters, as opposed to our precision maximization algorithm.

| Method | Alternative update? | Prec@1 | Prec@3 | Prec@5 |
|---|---|---|---|---|
| Wiki10-31K | | | | |
| XR-Linear | N/A | 84.14 | 72.85 | 64.09 |
| +Ours | No | 84.52 | 74.23 | 65.41 |
| +Ours | Yes | **84.57** | **74.55** | **65.59** |
| AmazonCat-13K | | | | |
| XR-Linear | N/A | 92.53 | 78.45 | 63.85 |
| +Ours | No | 90.63 | 78.12 | 63.86 |
| +Ours | Yes | **92.90** | **79.04** | **64.35** |
| Amazon-670K | | | | |
| XR-Linear | N/A | 44.17 | 39.14 | 35.38 |
| +Ours | No | 43.22 | 38.19 | 34.10 |
| +Ours | Yes | **44.82** | **39.93** | **36.34** |
| Amazon-3M | | | | |
| XR-Linear | N/A | 46.74 | 43.88 | 41.72 |
| +Ours | No | 46.95 | 44.24 | 42.18 |
| +Ours | Yes | **47.51** | **44.76** | **42.67** |

Table 4: Experiments with XR-Linear. Our method is called "XR-Linear+Ours", we also tested the finetune step detailed in Algorithm 1(L7). When joinly train option is disabled, the matcher $\mathcal{M}$ won't be updated despite the creation of new cluster map.

| Method | Extra training time | Extra inference time |
|---|---|---|
| Wiki10-31K | (baseline: $T_{trn} = 0\text{m}30\text{s}$, $T_{tst} = 0.7\text{ms/pts}$) | |
| +Ours ($\lambda = 1$) | 2.32× | 2.40× |
| +Ours ($\lambda = 2$) | 5.58× | 4.80× |
| AmazonCat-13K | (baseline: $T_{trn} = 1\text{m}37\text{s}$, $T_{tst} = 0.2\text{ms/pts}$) | |
| +Ours ($\lambda = 1$) | 3.05× | 0.71× |
| +Ours ($\lambda = 2$) | 4.54× | 1.59× |
| Amazon-670K | (baseline: $T_{trn} = 1\text{m}30\text{s}$, $T_{tst} = 0.4\text{ms/pts}$) | |
| Ours ($\lambda = 1$) | 0.47× | 0.07× |
| Ours ($\lambda = 2$) | 0.96× | 0.60× |
| Amazon-3M | (baseline: $T_{trn} = 13\text{m}49\text{s}$, $T_{tst} = 0.5\text{ms/pts}$) | |
| Ours ($\lambda = 1$) | 0.33× | 1.12× |
| Ours ($\lambda = 2$) | 1.63× | 2.51× |

Table 5: Extra training time and inference time (in ms/pts, which is millisecond per data point) of our method compared to the baseline XR-Linear. Our method posts negligible overhead for big model. For X-Transformer on Amazon-3M, we only have 1.1% training overhead and 21.1% inference overhead.

## 6 Conclusion

In this paper, we have proposed a simple way to disentangle the semantics reside in the labels of XMC problems. We have shown how to do this in an indirect way that builds overlapping label clusters. We proposed an optimization algorithm to solve both parts: the matcher model, the ranker and cluster assignments. In experiments, we tested under various cases and the results indicate that our method is exceptionally good when the label space is huge (such as Amazon-3M) or when the label contains many different meanings (such as the artificial data we created).

## Acknowledgement

This work is supported in part by NSF under IIS-1901527, IIS-2008173, IIS-2048280 and by Army Research Laboratory under agreement number W911NF-20-2-0158.

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
