# OpenReview forum: "Label Disentanglement in Partition-based Extreme Multilabel Classification"
_NeurIPS.cc/2021/Conference — NeurIPS 2021 Poster_

### Official Review · Reviewer_3RKd · 2021-07-14

**Rating:** 6
**Confidence:** 3

**Summary:**

This paper studies the extreme multi-label classification problem. Specifically, on top of the existing partition-based approaches that partition the label space into mutually exclusive clusters, the paper proposes a solution to relax this assumption by disentangle the multi-modal labels with non-exclusive clustering as an optimization approach. Evaluations are reported on four datasets against the existing baselines.

**Limitations And Societal Impact:**

The authors do not appear to address the limitation and the societal impact explicitly. One limitation is clearly that the complexity goes up when $\lambda$ increases even a little bit, which actually was verified in the evaluations even though just for $\lambda$ = 1 or 2.

**Main Review:**

The proposed method appears to technically make sense through the derivation and the evaluations. Novelty-wise, it is acceptable.

The major downside is the presentation, which is pretty rough with a lot of grammatical errors and typos.

Also in the reported evaluations (Table 3), why are the evaluation data using propensity scores far less impressive than those using precision scores? In fact, the paper states that “our proposed framework achieves new SOTA results on three out of four datasets”. This is misleading and confusing. More precisely, if precision is used, the proposed method achieves SOTA results for all the four datasets while if propensity is used, the proposed method only achieves SOTA results for two out of the four datasets.

---

I've read carefully the authors' rebuttal as well as the other reviews. Overall, though I see and agree to some of the concerns raised by the other reviewers, I like this work and thus stay with the original review.

**Time Spent Reviewing:**

5

---

> ### Author Response · Authors · 2021-08-10
> **Reply to 3RKd**
>
> We would like to thank you for all the comments and suggestions!
>
> > Q1: The major downside is the presentation, which is pretty rough with a lot of grammatical errors and typos.
>
> A1: We are very sorry about that. We will carefully proofread our drafts and correct the typos as much as we can.
>
>
> > Q2: The paper states that “our proposed framework achieves new SOTA results on three out of four datasets”. This is misleading and confusing.
>
> A2: Our claim that we set the new SOTA is primarily based on precision@k. Moreover, our PSP@k beats the other methods on the two of the larger datasets. But we will correct our argument in the revision.

---

### Official Review · Reviewer_h4mU · 2021-07-15

**Rating:** 7
**Confidence:** 4

**Summary:**

The authors propose a new method for partitioning labels space that can be used with a family of currently popular partition-based/label tree-based extreme multi-label classifiers, which organize labels in the form of the tree. The proposed method introduces redundancy of labels in the different label clusters, which are mutually exclusive to the clusters in many methods. The problem of label assignment is formulated as maximization of matcher's (part of classifier predicting clusters for labels) precision, which is optimized by adding new labels to the cluster after initial training with clusters with mutually exclusive labels. The new partitioning method is applied to XL-Linear and X-TRANSFORMERS in the experimental study, resulting in a steady improvement in predictive performance on four popular benchmark datasets.

**Ethical Concerns:**

No ethical issues were found.

**Limitations And Societal Impact:**

The limitations were addressed and no issues with negative social impact were found.

**Main Review:**

Strengths:
+ The paper has a clear motivation and introduced technique is applicable to a wide range of extreme classification label tree-based methods.
+ The organization of this paper is good, and it's easy to read and understand.
+ Solid empirical study on the synthetic dataset by grouping many labels into one, performance evaluation, and comparison with random baseline.
+ New method combined with X-TRANSFORMERS models (ensemble of 9 of them) leads to SOTA results on standard benchmarks while only slightly increasing computational complexity.

Weaknesses:
- The scope of the contribution is a bit limited, the idea and the results are not surprising. It's a small iterative improvement to the formula of label tree-based methods.
- Empirical study uses only four datasets, where Amazon-670K and Amazon-3M are datasets representing exactly the same problem. I lack comparison, at least on the popular Wikipedia-500K benchmark.

Other comments:
* I think most of the tail labels are unimodal, and redundancy just improves their chances to be in one of the clusters returned by the matcher (what is basically the objective in the proposed approach). Since there is no penalty for assigning labels to up to additional lambda clusters in the optimization objective, I believe most of the labels, if not all of them, end up being redundant, even labels with only one data point in the train set? Maybe different values of lambda should be used for different labels?
* There is no comment on the value of lambda in the synthetic experiment. It was equal to lambda, right?
* Which dataset is the source for the examples in Table 2?
---
Edit after the authors' response: I believe that the authors did a good job providing additional experiments and discussing my and other reviewers' comments. I'm raising my rating to 7.

**Time Spent Reviewing:**

6

---

> ### Author Response · Authors · 2021-08-10
> **Reply to h4mU**
>
> Thanks for your comments and suggestions!
>
> > A1: Empirical study uses only four datasets, where Amazon-670K and Amazon-3M are datasets representing exactly the same problem. I lack comparison, at least on the popular Wikipedia-500K benchmark.
>
> Q1: We have conducted experiments on the Wikipedia-500K dataset on XR-Linear. The results in the following table also demonstrate improvements over the baseline. We will add this to the revision.
>
> | Lambda   |    Prec@1 |   Prec@3  |  Prec@5  |  Recall@1 |  Recall@3  |  Recall@5 |
> | ------------- | ------------- | ------------- | -------------- | -------------| ---------- | ----------------- |
> | 0 (baseline) | 66.63    |  47.63    |      37.16      |  21.67       |   39.46   |    47.53 |
> | 1            | 66.79        |  47.76     | 37.23          | 21.68         | 39.52     |    47.60  |
> | 2            | 68.06        |   48.85     | 38.17          | 22.20        | 40.70      |   49.15  |
>
>
> > A2:  I believe most of the labels, if not all of them, end up being redundant, even labels with only one data point in the train set? Maybe different values of lambda should be used for different labels?
>
> Q2: We believe this is the same question as Reviewer Rv3e (we answered in Q5/A5 section). This is an interesting idea that we had already tried but not presented here. We created another model that duplicates more head labels and fewer tail labels. This is done by releasing the constraint in Eq(9). After doing that, most duplications are from head labels because the unconstrained projection $Proj(Y^T M)$ is simply sorting the entries $(Y^T M)_{ij}$ in descending order, where head labels are selected in higher priority.
>
>  |                           | P@1 |   P@3  |  P@5  |  R@1 |   R@3 |   R@5 |
> | -------------------- | --------- | -------- | --------  | ----- | ---------- | --------- |
> | Unbalanced (more head, less tail) |   82.66 | 68.87 | 57.74 | 16.68 | 40.92 | 56.13 |
> | Our method(balanced)  |   82.74 | 69.81 | 58.48 | 16.74 | 41.47 | 56.83 |
> **Dataset: Eurlex-4k**
>
> > Q3: There is no comment on the value of lambda in the synthetic experiment. It was equal to lambda, right?
>
> A3: Yes exactly. In this experiment, the lambda equals the number of real labels in the synthetic labels.
>
> > Q4: Which dataset is the source for the examples in Table 2?
>
> A4: These are examples from the Amazon-670K dataset.

---

> > ### Comment · Reviewer_h4mU · 2021-09-01
> > **Raising my score**
> >
> > Dear Authors,
> >
> > thank you for providing additional experiments and answering my and other reviewers' questions. Your responses strengthen my opinion that this work deserves to be accepted at the conference, so I would l would like to raise my score to 7.
> >
> > Still, it would be nice to see results for Wiki-500 for your method combined with X-Transformers, since this will help with feature comparisons and because you just retrain tree-part obtaining these results shouldn't be difficult.

---

### Official Review · Reviewer_2ebq · 2021-07-16

**Rating:** 4
**Confidence:** 4

**Summary:**

In extreme Multi-label classification (XML) a large range of methods based their prediction on partitioning whether the input space or label space.  The paper addresses this partitioning phase, proposing a "disentangled partitioning", allowing overlapping clustering of labels. The proposed model consists to use two matrices, the first one M corresponding to the classification of the examples to each cluster (the matcher) and the second one C indicating the assignment of the labels to each cluster.  The goal is to optimize those matrices ensuring that  MC is close to the label matrix under constraints to ensure that each label is present up to lambda time in different clusters (the disentanglement factor). After a formal analysis of the problem, the authors propose a simple algorithm to optimize the matrix C, using existing approaches to learn the matcher. To validate the approach authors first propose an experiment merging artificially labels for clustering. The second series of experiments evaluate the performances wrt SOTA algorithms and varying the disentanglement parameter.


**Main Review:**

Originality: The subject is not particularly new, the formalization uses standard tools (binary matrices), the proofs are quite straightforward.

Clarity: The paper can be quite difficult to read and is not always clear particularly in the experimental section when we are not sure of the backend used.  For instance, Table 5 is not clear enough the caption mentioning X-transformer and XR-linear. Another example, the constraint lambda is claimed to be an upper limit of the number of times that a label appears in clusters, however, equation 10 seems to select each label exactly lambda times.

Quality:
* The first major flaw of the paper concerns the adequation of the thesis of the paper and the experiments conducted. The authors claim to provide a model allowing to disentangle the semantics, but no experiments are showing exactly this fact. Experiments show better precision in different settings, but I am not convinced that the performances are due to disentanglement rather than redundancy of the labels.  There is no guarantee that for a given label its two assigned clusters differ semantically. Additional results showing the distance between clusters sharing some labels or with some kind of orthogonality measure between overlapping clusters should have been provided to support the author's thesis.

* Concerning the core experiment Table 3, most of the table is not informative. The proposed approach uses other algorithms as backend, thus the only interesting lines are those concerning the backend algorithm (x-transformer) and the proposed plugin approach to measure the improvement of the proposal.  Further experiments should have been conducted with other usual backends (AnnexML, DiSmec at least) to fully assess the benefits of the approach. Moreover, to the best of my knowledge, X-transformer and X-linear (the two studied backends) do not use ensembling in the presented results. It would have been fairer to compare to results using ensembling allowing thus a similar expressive power wrt to the redundancy of the labels.

* Concerning the proposed solution to problem 9, the solution seems too naive: it consists to assign to each label the lambda clusters where it is the most present. I am not sure that this solution does not lead to very correlated clusters (see above) and leads effectively to semantic disentanglement. The approximation made by Eq 8. seems too wide to lead to efficient solutions.

Significance: The addressed problem is interesting but the proposed approach seems too naive to work in generic cases. More experiments are needed to assess the impact of the plugin and to show that it provides significant improvement to semantic disentanglement.

**Time Spent Reviewing:**

6

---

> ### Author Response · Authors · 2021-08-10
> **Reply to 2ebq**
>
> We thank Reviewer 2ebq for the insightful comments and suggestions. We have addressed each of your questions below.
>
> > A1: No experimental support showing that it is disentangling the semantics plays the role, rather than duplicating the labels.
>
> Q1: Thanks for your suggestion! We designed a new experiment to analyze whether our approach can disentangle the mixed semantics of labels. This experiment is an extension of our existing one detailed in Section 5.1, where we created pseudo multi-modal labels by grouping several labels on real-world XMC datasets such as Amazon-670k.
>
> Let’s start with an example pseudo label: $\bar{L}=\langle \text{car accessories}\rangle + \langle \text{body wash}\rangle$, which is merged from two different labels $L_1=\langle \text{car accessories}\rangle$ and $L_2=\langle \text{body wash}\rangle$. A perfect semantic disentanglement of $\bar{L}$ is the partition of positive instances of $\bar{L}=[L_1, L_2]$ (defined as $Pos(\bar{L})$ = {$x_i\ |\ y_{i,1}=1\text{ or } y_{i,2}=1$}) into the following two disjoint subsets:  $S^*_1\cup S^*_2$ where $S^*_1$ contains all positive instances of  $L_1$ and $S^*_2$ contains all positive instances of $L_2$.
>
> For the same pseudo label $\bar{L}$, our proposed method can also create a partition on the same positive instance set $Pos(\bar{L})$. Let’s say our method assign $\bar{L}$ into two clusters $A$ and $B$. Then the corresponding partition would be $S'_1$ = { $x_i | \text{ score of }c_A > \text{score of } c_B$ }; and $S'_2=Pos(\bar{L}) - S'_1$. Note that $S’_1$ are the samples in $Pos(\bar{L})$ that are matched to cluster $A$ and $S'_2$ are the rest of the samples in $Pos(\bar{L})$.
>
> We can quantitatively measure the distance between two different partitions $S’$ = { $S'_1, S'_2$ }
>  vs $S^*$ = { $S^*_1, S^*_2$ } using the notion of Variation of Information (VI) [1,2], which has been
> widely-used in the data mining community with strong theoretical justification.
>
> We then repeat the same process above to all synthetic labels $\langle \text{label 1}\rangle + \langle \text{label2}\rangle$ with two duplicates $L_1 / L_2$. Then compare the alignment of the following four sets:
>
> - $S^*_1$: defined as all positive instances of $\text{label1}$.
> - $S^*_2$: defined as all positive instances of $\text{label2}$.
> - $S'_1$: containing all instances such that the matcher assigns higher scores to $L_1$ over $L_2$.
> - $S’_2$: containing all instances such that the matcher assigns higher scores to $L_2$ over $L_1$.
>
> The degree of alignment is formally measured by the variation of information (VI): two sets $S^*$ = { $S^*_1, S^*_2$ } and $S’$ = { $S’_1, S’_2$ }, the VI is computed as
>
>  $VI = - \sum_{i,j}  r_{ij} * (\log \frac{r_{ij}}{p_i} + \log \frac{r_{ij}}{q_j})$
>
> Where $r_{ij} = \frac{|S^*_i \cap S’_j|}{n}$, $p_i = \frac{|S^*_i|}{n}$ , $q_j = \frac{|S’_j|}{n}$ and $n = |S^*_1| + |S^*_2| = |S’_1| + |S’_2|$
>
> **We plot the histogram of VI and compare our disentanglement with random duplication. The result is here https://postimg.cc/bZJ8hGFd (dataset is Amazon-670k).**
>
> From this figure, we can observe that the random baseline (with no effective semantic disentanglement) has the variation of information values concentrated around 1.3; while our method has a wider distribution between 0 and 1. **Notably, there are more than 20 synthetic labels been perfectly disentangled with 100% accuracy (the VI distance metric is zero in this case).**
>
> [1]: Arabie, P.; Boorman, S. A. (1973). Multidimensional scaling of measures of distance between partitions. Journal of Mathematical Psychology. 10 (2): 148–203.
>
> [2]: Meilă, M. (2003). Comparing clusterings by the variation of information. In Learning theory and kernel machines (pp. 173-187). Springer, Berlin, Heidelberg.
>
>
> > Q2: Further experiments should have been conducted with other usual backends (AnnexML, DiSmec at least).
>
> A2:  Our paper only focuses on tree-based XMC models, so we conducted experiments on two represented ones (a linear-based model and a transformer-based model). AnnexML is an ANN-based XMC model and DiSmec is based on plain linear one-versus-all.
>
> > Q3: The approximation made by Eq 8. seems too wide to lead to efficient solutions.
>
> A3: The original problem is an integer programming problem and we showed it’s NP-complete (Theorem 1). Without the approximation, it would be intractable in our problem scale. So we decided to relax it to a linear programming problem.

---

> ### Author Response · Authors · 2021-08-30
> **Any further comments?**
>
> Dear reviewer,
>
> We really appreciate your valuable comments. Since the discussion period is ending soon, we would like to hear your thoughts after reviewing our additional experiments in the previous post.
>
> In summary, our new experiment verified your suggestion that labels are semantically disentangled throughout the duplication process. *Doing so quantitatively is a non-trivial thing*. We first create a synthetic dataset by combining two different labels and then see if our model separates them by duplicating this synthetic label to another cluster. The variational information (VI) metric is a standard way to measure how well the newly created two labels align with the ground truth labels. Therefore, we compare the VI between our method with the baseline that duplicates the label unstrategically. The result (https://postimg.cc/bZJ8hGFd) exhibits that our approach tends to have a much lower VI, meaning that the model disentangles the label space in a semantically meaningful way.
>
> Once again, we are grateful to the reviewer for providing this insightful direction. We are still looking forward to more discussions to help you evaluate our paper.
>
> Sincerely,
>
> Authors of this submission

---

### Official Review · Reviewer_Rv3e · 2021-07-18

**Rating:** 7
**Confidence:** 4

**Summary:**

Label partitioning approaches are well-explored in Extreme Classification (XC) literature whose core idea is to cluster semantically similar labels together and then to sample the negative labels only from a small number of relevant clusters. Although this has been shown to significantly speed up the training and prediction routines, most existing partitioners simplistically assume that each label belongs to exactly one cluster. This could degrade prediction accuracies since labels in XC can often be multi-modal with multiple associated semantics. This paper addresses this issue by relaxing the single cluster assumption and thus allowing each label to instead end up in \lambda different clusters. This is achieved though a novel cluster assignment algorithm which is shown to be theoretically well-defined and leads to small but consistent accuracy gains. The proposed algorithm has the added advantage of being plug-and-play, i.e. it can be used with many matching and ranking models based on label partitioning. Enough experiments have been conducted on simulated and real datasets to demonstrate the accuracy gains and discuss the associated increases in costs.

**Limitations And Societal Impact:**

Yes limitations have been addressed. Negative societal impact is not applicable.

**Main Review:**

The paper introduces a neat and novel idea in label partitioning for XC which is easily adoptable within many models. As demonstrated in experiments, the approach could lead to small (< 1%) accuracy gains in different models and across datasets despite a modest 2x increment in training and prediction costs. Due to this, the paper could have decent amount of impact in XC applications.

The intuitions for the core ideas are well presented and easy to understand and appreciate. The idea of overlapping label partitions is novel and worth more exploration in XC. The proposed partitioning algorithm is novel and the theoretical contributions are simple but well motivated. The empirical results address most of the pragmatic concerns. In summary, the technical contributions are sound and sufficient. The paper is mostly well-written and easy to follow.

There are a few drawbacks of this paper:
(1) Accuracy gains of 1% are not that impressive especially due to associated increases in cost
(2) Some obvious questions which arise are left unresolved:
    a) What is the effect on accuracy and convergence due to multiple iterations of model training and cluster mapping? Does it continuously improve performance or hurt it after some iterations. In general, what is the prescription here?
    b) Although it is clear that overlapping clusters lead to accuracy gains, it is however unclear whether these gains are due to improvements in matching module (due to semantically well separated clusters) or  in the ranking module (due to more and diverse negative labels to learn from)?
(3) Minor issues:
    a) There has been some prior work in XC on co-optimizing clustering and XML objectives, e.g. [1,2]. Although these are point-partitioning approaches, they are worth mentioning for the sake of completeness
    b) Line 167: The second constraint is not intended to mitigate the case of degenerated clusters as mentioned here. In fact, as noted in the Line 199, cluster imbalance is avoided due to good initialization from balanced k=2 means clustering
    c) Head labels might be semantically more complex than tail labels. Does it help to put head labels in more clusters than tail ones?
    d) Why is averaging duplicate label scores better than other alternatives, e.g. taking max over them ?
    e)  line 53: one-vs-reset -> one-vs-rest
        lines 73-74: in my understanding, SLICE graph construction is not unsupervised or agnostic to data distribution since it uses centroid classifiers as label representations during graph construction
        line 78: imbalance -> imbalanced

[1] Simultaneous Learning of Trees and Representations for Extreme Classification and Density Estimation
Yacine Jernite, Anna Choromanska, David Sontag
[2] Choromanska, A. and Langford, J. Logarithmic time online
multiclass prediction

**Time Spent Reviewing:**

10

---

> ### Author Response · Authors · 2021-08-10
> **Reply to Rv3e**
>
> We thank Reviewer Rv3e for the insightful comments and suggestions. We have addressed each of your questions below.
>
> > Q1: What is the effect on accuracy and convergence due to multiple iterations of model training and cluster mapping?
>
> A1: For the datasets we tested, there is a very small benefit of going through more than two cycles comparing with our current version. The accuracy gain is not worth the extra computational time, so we only did k-means cluster -> matcher -> ranker -> overlap cluster -> new matcher -> new ranker.
>
> > Q2: Unclear whether these gains are due to improvements in matching module.
>
> A2: We made more detailed profiling of which component contributes more to the accuracy gain. The results indicate that both the matcher and the ranker benefit from overlapping clusters. The profiling is conduct in the following steps:
>
> 1. To check the improvement of the matcher, we replace the ranker with <golden ranker>, meaning the ranker does not make any mistake (by looking at the ground truth Y). So all the errors are originated from the matcher. We compare the metrics with the baseline model:
>
>  |                           | P@1 |   P@3  |  P@5  |  R@1 |   R@3 |   R@5 |
> | -------------------- | --------- | -------- | --------  | ----- | ---------- | --------- |
> |Our matcher + Golden Ranker |   99.87 | 98.10 | 89.02 | 20.50 | 59.16 | 86.07 |
> |Baseline matcher + Golden Ranker       | 99.79 | 97.51 | 86.77 | 20.48 | 58.71 | 83.83 |
> **Dataset: Eurlex-4k**
>
>
> 2. To check the improvement of the ranker, we replace the matcher in our new model with the one in the baseline model. Given that the two matchers are identical, the difference of errors are purely from ranker:
>  |                           | P@1 |   P@3  |  P@5  |  R@1 |   R@3 |   R@5 |
> | -------------------- | --------- | -------- | --------  | ----- | ---------- | --------- |
> | Our Ranker     |  82.74  | 69.81  | 58.48  | 16.74 | 41.47 | 56.83 |
>  | Baseline Ranker | 82.48 | 68.83 | 57.61 | 16.65 | 40.85 | 55.98 |
> **Dataset: Eurlex-4k**
>
> > Q3: Prior work in XC on co-optimizing clustering and XML objectives, e.g. [1,2]
>
> A3: Thanks for introducing these papers! We checked out the papers, both [1,2] jointly optimizes the tree structure in a different way so that the label tree is balanced and label distribution is pure. Apart from the differences you mentioned, our tree structures are constructed in one pass with hierarchical k-means.
>
> [1] Y. Jernite et al., Simultaneous Learning of Trees and Representations for Extreme Classification and Density Estimation. ICML 2017
>
> [2] Choromanska, A. and Langford, J., Logarithmic time online multiclass prediction. NeurIPS 2015
>
>
> > Q4: Line 167: The second constraint is not intended to mitigate the case of degenerated clusters as mentioned here.
>
> A4: Thanks for pointing out this imprecise description. We actually meant that lambda is a constraint that limits the number of duplicates for any label so that the number of duplicates won’t be highly imbalanced among **labels**. The cluster imbalance problem is mainly addressed by a good initialization.
>
> > Q5: Head labels might be semantically more complex than tail labels. Does it help to put head labels in more clusters than tail ones?
>
> A5: This is an interesting idea that we had already tried but not presented here. We created another model that duplicates more head labels and fewer tail labels. This is done by releasing the constraint in Eq(9). After doing that, most duplications are from head labels because the unconstrained projection $Proj(Y^T M)$ is simply sorting the entries $(Y^T M)_{ij}$ in descending order, where head labels are selected in higher priority.
>
>  |                           | P@1 |   P@3  |  P@5  |  R@1 |   R@3 |   R@5 |
> | -------------------- | --------- | -------- | --------  | ----- | ---------- | --------- |
> | Unbalanced (more head, less tail) |   82.66 | 68.87 | 57.74 | 16.68 | 40.92 | 56.13 |
> | Our method(balanced)  |   82.74 | 69.81 | 58.48 | 16.74 | 41.47 | 56.83 |
> **Dataset: Eurlex-4k**
>
> As we can see, by removing the constraint, there is a small accuracy drop in both accuracy and recall. However, there may be other better ways to non-uniformly duplicate labels, which is an interesting future direction.
>
> > Q6: Why is averaging duplicate label scores better than other alternatives, e.g. taking max over them?
>
> A6: This is another interesting exploration that we tried but not presented in the paper.
> Due to the noise in the score estimation, the max aggregator tends to overestimate the true score, and the bias increases as more and more duplicates are inserted. Meanwhile, the mean aggregator has a more stable performance (e.g., P@1 around [82.5, 83.0]) as our paper shows. Here is the plot: https://postimg.cc/QKbC4vCP

---

### Decision · Program_Chairs · 2021-09-27

**Decision:**

Accept (Poster)

**Comment:**

The authors propose a simple plug-in technique to improve the performance of label-partition based extreme classification algorithm. The improvements in performance are small but consistent.The paper is clear and easy to follow.  On the downside, the contribution is a bit limited and the method does incur a computational overhead, which might be significant for more efficient XC algorithms. The theoretical results in the paper are rather obvious, and under unrealistic assumptions, but they do help motivate the method.  The paper is also missing a comparison on the Wikipedia 500K dataset which is a standard dataset in the extreme classification.